# Somatic recombination underlies frequent revertant mosaicism in loricrin keratoderma

Shotaro Suzuki[1], Toshifumi Nomura[1] , Toshinari Miyauchi[1], Masae Takeda[1], Yasuyuki Fujita[1], Wataru Nishie[1], Masashi Akiyama[2], Akemi Ishida-Yamamoto[3], Hiroshi Shimizu[1]

**Revertant mosaicism is a phenomenon in which pathogenic mutations are rescued by somatic events, representing a form of natural gene therapy. Here, we report on the first evidence for revertant mosaicism in loricrin keratoderma (LK), an autosomal dominant form of ichthyosis caused by mutations in *LOR* on 1q21.3. We identified two unrelated LK families exhibiting dozens of previously unreported white spots, which increased in both number and size with age. Biopsies of these spots revealed that they had normal histology and that causal *LOR* mutations were lost. Notably, dense single nucleotide polymorphism mapping identified independent copy-neutral loss-of-heterozygosity events on chromosome 1q extending from regions centromeric to *LOR* to the telomere in all investigated spots, suggesting that somatic recombination represents a common reversion mechanism in LK. Furthermore, we demonstrated that reversion of *LOR* mutations confers a growth advantage to cells in vitro, but the clinically limited size of revertant spots suggests the existence of mechanisms constraining revertant clone expansion. Nevertheless, the identification of revertant mosaicism in LK might pave the way for revertant therapy for this intractable disease.**

## Introduction

The epidermis, the outermost layer of the skin, is a tissue with a high turnover rate that is constantly being replenished by skin-residing stem cells (SCs) (Gonzales & Fuchs, 2017). Epidermal SCs self-renew and generate more-differentiated epidermal cells called keratinocytes throughout life, inevitably acquiring mutations that result from both environmental factors and DNA replication errors with age. Although most of these mutations do not have a noticeable effect, some are deleterious, and the age-related accumulation of somatic mutations in SCs potentially contributes to carcinogenesis and the decline of tissue function (Adams et al, 2015; Siudeja et al, 2015; Blokzijl et al, 2016). In theory, such genetic alterations,

including point mutations, gene conversion, and recombination, can also lead to natural correction of pathogenic mutations at the somatic-cell level in genetic disorders. Somatic reversion of a mutant phenotype is referred to as revertant mosaicism and results from the correction of a disease-causing mutation in a somatic cell followed by the survival and clonal expansion of the revertant cell (Jonkman & Pasmooij, 2009). However, this clinically important "natural gene therapy" phenomenon has been documented in only ~30 diseases, including seven genodermatoses caused by mutations in *COL7A1* (recessive dystrophic epidermolysis bullosa [EB, Mendelian inheritance in man (MIM): 226600]), *COL17A1* or *LAMB3* (junctional EB [MIM: 226650]), *KRT14* (EB simplex [MIM: 131900]), *FERMT1* (Kindler syndrome [MIM: 173650]), *KRT10* or *KRT1* (ichthyosis with confetti [IWC, MIM: 609165]), and *GJB2* (keratitis-ichthyosis-deafness syndrome [MIM: 148210]) (Jonkman & Pasmooij, 2009; Gudmundsson et al, 2017; Lim et al, 2017; van den Akker et al, 2018). Although mutation-free induced pluripotent SCs established from revertant junctional EB keratinocytes are already on the way to clinical translation (Umegaki-Arao et al, 2014), the feasibility and generality of this therapeutic approach appear to be restricted by the currently limited number of diseases that are known to exhibit revertant mosaicism, warranting expansion of the repertoire of such diseases.

Loricrin keratoderma (LK [MIM: 604117]) is a rare autosomal dominant skin disorder that results from gain-of-function mutations in *LOR* (MIM: 152445), which encodes loricrin, on 1q21.3 (Ishida-Yamamoto et al, 1997; Ishida-Yamamoto, 2003). The late epidermal differentiation marker loricrin is the major component of the cornified envelope, a highly insoluble and robust structure that is formed beneath the cell membrane of keratinocytes during terminal differentiation and provides the vital physical barrier to the skin (Ishida-Yamamoto, 2003). LK is clinically characterized by defective skin barrier phenotypes, such as palmoplantar keratoderma (thickening of the skin on the palms and soles; PPK) and generalized ichthyosis (dry, thickened, and scaly skin), with or without varying degrees of constricted digits, erythematous plaques, and/or erythroderma (generalized red skin) (Maestrini et al, 1996; Ishida-Yamamoto et al, 1997; Ishida-Yamamoto, 2003; Pohler et al, 2015). Notably, all *LOR* mutations reported to date

[1]Department of Dermatology, Hokkaido University Graduate School of Medicine, Sapporo, Japan  [2]Department of Dermatology, Nagoya University Graduate School of Medicine, Nagoya, Japan  [3]Department of Dermatology, Asahikawa Medical University, Asahikawa, Japan

Correspondence: nomura@huhp.hokudai.ac.jp

cause a frameshift and replace the glycine-rich C-terminus of the 315-amino-acid wild-type loricrin protein with a highly arginine-rich nuclear localization signal sequence (Maestrini et al, 1996; Ishida-Yamamoto et al, 1997; Ishida-Yamamoto, 2003; Pohler et al, 2015; Khalil et al, 2017), leading to the accumulation of mutant loricrin in the nucleolus of affected keratinocytes (Ishida-Yamamoto, 2003). Although the precise mode of action of mutant loricrin in the pathogenesis of LK is not fully understood, its elimination is crucial for curing the disease.

Here, we describe two families with LK exhibiting progressive development of revertant skin areas in which causal mutations were corrected. Of particular note, the reversion mechanism in six investigated revertant epidermis samples was somatic recombination. The frequent occurrence of mutation-reversion events via somatic recombination was a remarkable feature of these LK families.

## Results

### Identification of revertant skin spots in two unrelated LK patients

In family 1, there were three affected individuals—the proband, a 51-yr-old man born to unaffected parents, and his two daughters, aged 12 and 9 (Fig 1A)—exhibiting the hallmark features of LK (i.e., generalized ichthyosis, PPK, constricted digits, and/or well-demarcated erythematous plaques; Figs 1B–D and S1). Histology of the lesional skin sampled from the proband revealed hyper-keratosis, parakeratosis (retained nuclei in the stratum corneum [the outermost layer of the epidermis]), prominent keratohyalin granules, and acanthosis, all of which are characteristic features of LK (Fig 1E and F). Whole-exome and Sanger sequencing in three affected and two unaffected individuals in the family revealed that the affected individuals were all heterozygous for a 1-bp insertion mutation in LOR, c.545_546insG (p.Gly183ArgfsTer153) (Fig 1G), which had previously been reported to cause LK (Song et al, 2008), further verifying the diagnosis of LK. By contrast, the unaffected individuals were both wild-type for the mutation. Notably, we found that erythematous skin of the proband was interrupted by dozens of whitish, normal-appearing skin patches on his trunk and extremities (Fig 1H and I). The patches, which were up to 10 mm in diameter, were slightly depressed from the surrounding affected skin (Fig 1I). Dermoscopy, a noninvasive in vivo technique that allows detailed examination of skin lesions, revealed that the patches exhibited no scaling (Fig 1J), suggesting the possibility that they represented revertant skin areas.

To test this possibility, we sampled skin from one of the whitish patches, which was then subjected to histological and genetic analyses. Notably, the examined patch showed normal histology, with a normal basket-weave stratum corneum (Fig 1K). Immuno-histochemical analysis revealed that mutant loricrin mislocalized to the nuclei of the affected keratinocytes (Fig 2A–K), which reflects the fact that it contains an arginine-rich nuclear localization signal motif (Fig 2L). In contrast, mutant loricrin was not detected in the keratinocytes sampled from the normal-appearing patch (Fig 2K). We next examined the LOR genotype in genomic DNA samples extracted separately from the epidermis and dermis of the normal-appearing patch. Notably, the mutation was absent in the epidermis but remained in the dermis (Fig 1L). Thus, we confirmed that the clinically and histologically verified reversal of symptoms was due to the somatic reversion of the pathogenic mutation in the epidermis.

To clarify whether his other normal-appearing skin patches also represent revertant spots, we analysed three additional patches. Notably, they also showed normal histology and somatic reversion of the pathogenic mutation in the epidermis (Figs S2 and S3). Thus, we confirmed the multiple occurrence of somatic reversion of the LOR mutation in the proband. To the best of our knowledge, however, revertant mosaicism has never been noted in previously reported LK cases.

To test the generality of revertant mosaicism in LK, we analysed a second family with the disease. In family 2, there were two alive affected individuals whose detailed clinicohistological and genetic features had been published elsewhere (Ishida-Yamamoto et al, 1997 and Fig 3A). Briefly, they exhibited generalized ichthyosis, PPK, constricted fingers, and widespread erythematous plaques (Fig 3B–E) and carried a heterozygous mutation, c.664_665insC, in LOR (Ishida-Yamamoto et al, 1997). The mutation results in a frameshift and delayed termination (p.Gln222ProfsTer116), yielding a C-terminally extended arginine-rich mutant loricrin protein (Fig 2L). Of particular note, careful re-evaluation of the 58-yr-old proband's skin phenotypes revealed that her erythematous skin was also interrupted by dozens of non-scaly, whitish skin spots, which were up to 10 mm in size and were noted almost exclusively on the lower extremities (Fig 3D and E). To validate whether they represent revertant spots, two of the clinically normal-appearing patches and the lesional skin of her left posterior thigh were sampled for histological and genetic analyses. Histology of the lesional skin showed marked hyper-keratosis with parakeratosis and hypergranulosis due to the mutation (Fig 3F–H). In contrast, the two normal-appearing patches displayed normal epidermal architecture (Fig 3I–L). Sanger sequencing revealed that the disease-causing mutation had almost reverted in the epidermis, but not in the dermis, of the two histologically cured patches (Fig 3M and N). Taken together, we concluded that the clinically and histologically verified reversal of her skin phenotypes resulted from somatic reversion of the disease-causing mutation to wild-type.

### Age as a key factor for the development of clinically recognizable revertant spots

In family 1, according to the proband's personal statement, the normal patches had appeared at age 20 and had gradually increased in both number and size over decades (Fig S4). In family 2, the photographic record revealed that the proband had not exhibited these spots at the age of 27 (Fig S5), indicating that they had become detectable after that age. Furthermore, the proband's two affected daughters, aged 12 and 9, in family 1 did not exhibit normal-appearing patches (Fig S1), whereas the affected father of the proband in family 2 showed multiple clinically reverted skin spots when he was 57 (Fig S6). These findings suggest that LK patients may develop revertant spots for the first time in their

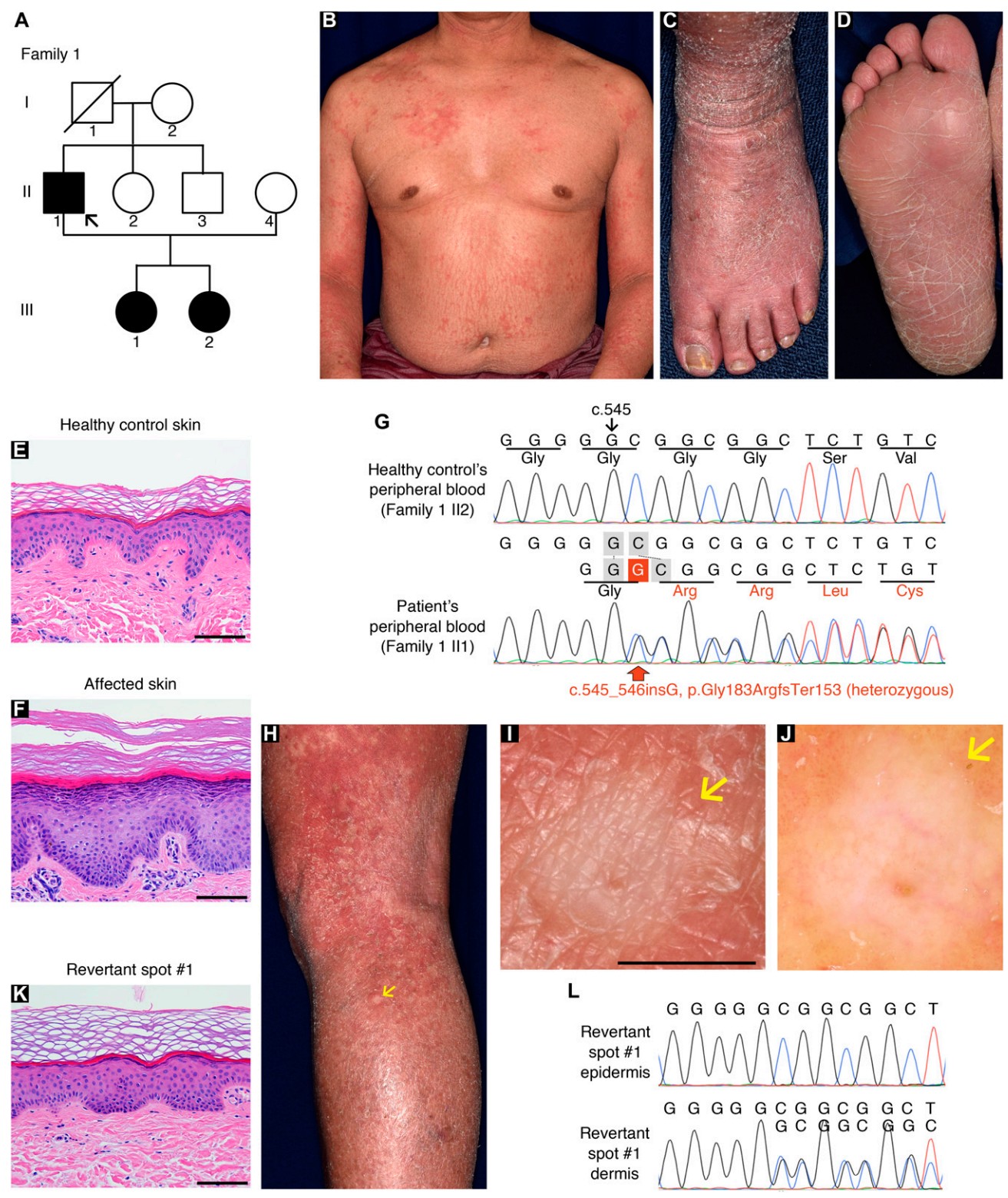

**Figure 1. Clinical, histological, and genetic features of the proband in LK family 1.**
**(A)** Pedigree. The arrow indicates the proband. **(B, C)** Generalized ichthyosis and irregularly shaped erythematous plaques. **(D)** His plantar skin was diffusely hyperkeratotic. **(E, F)** Compared with healthy control skin (E), the patient's affected epidermis (F) showed hyperkeratosis, parakeratosis, prominent keratohyalin granules, and acanthosis. **(G)** The proband was heterozygous for a 1-bp insertion mutation, c.545_546insG (p.Gly183ArgfsTer153). **(H–L)** Gross (H, I), dermoscopic (J), histological (K), and genetic (L) features of the revertant spot on his left leg. Note that no scales were observed on the revertant spot (arrows). **(K)** The revertant spot showed histological normalization. **(L)** The frameshift mutation was lost in the revertant epidermis. Scale bars, 100 μm (E, F, and K) and 5 mm (I).

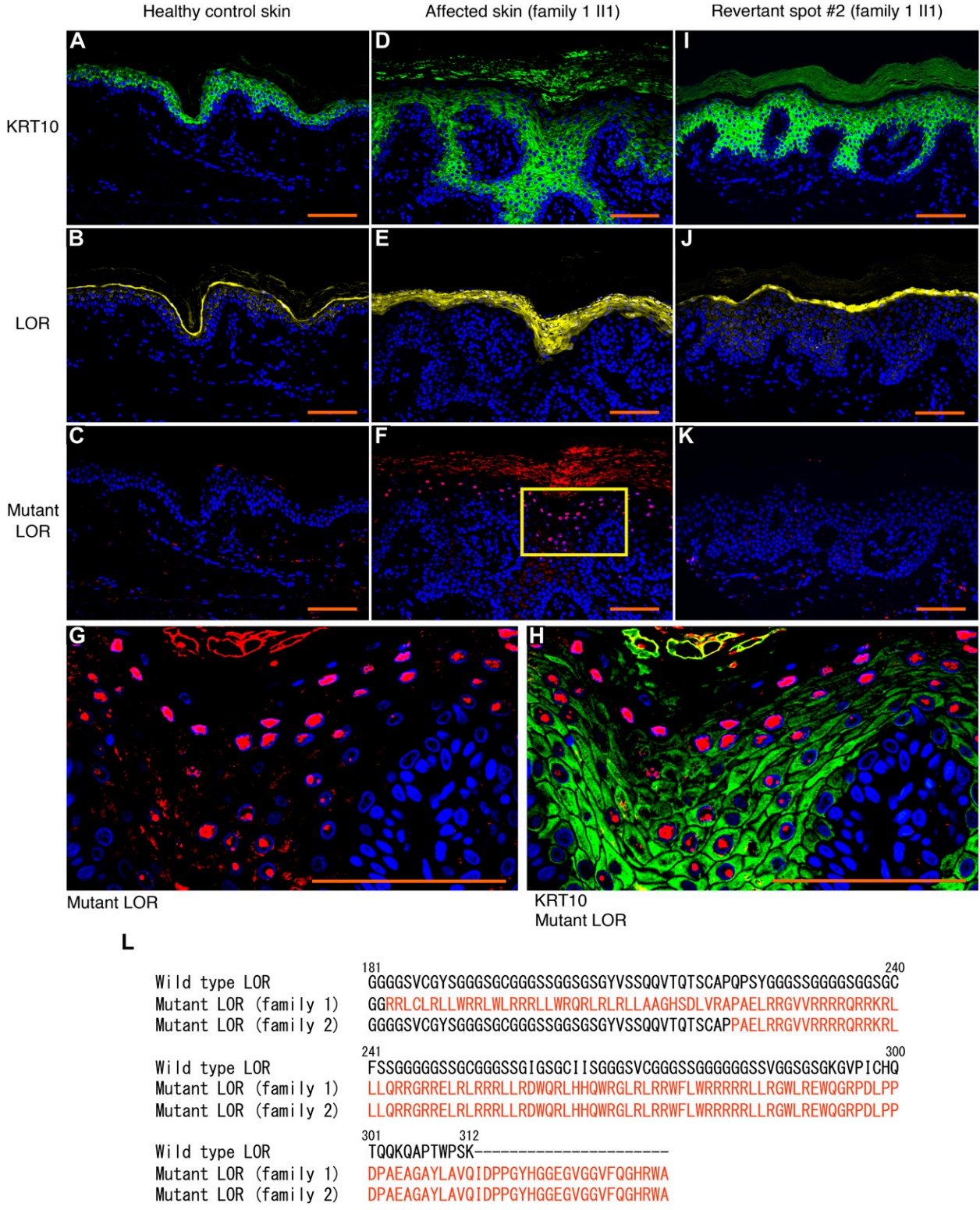

**Figure 2.  Immunolocalization of mutant LOR and its C-terminal amino acid sequence.**
**(A–C)** In healthy control epidermis, KRT10 and wild-type LOR were expressed by suprabasal keratinocytes and more-differentiated keratinocytes, respectively. No mutant LOR was detected. **(D–H)** Immunolocalization of KRT10 and wild-type LOR revealed thicker stratum spinosum and granulosum in the affected epidermis than in the control. Mutant LOR accumulated in the nuclei of keratinocytes in the upper epidermis. Higher magnification of the square in (F) is shown in (G). Merged image of anti-KRT10 and anti-mutant LOR was also shown (H). **(I–K)** Immunolocalization of KRT10, wild-type LOR and mutant LOR was normalized in the revertant epidermis. Scale bars, 100 µm. **(L)** The C-terminal amino acid sequences of the wild-type and mutant LOR. We identified two frameshift mutations in *LOR*, namely, c.545_546insG

twenties or even later, with the number and size of such spots increasing.

### Reversion of the *LOR* mutations results from somatic recombination

To address the mechanisms underlying the reversion of the pathogenic mutations in LK, we next performed high-density, genome-wide genotyping. For this analysis, genomic DNA separately extracted from the epidermis of the aforementioned six revertant and two affected patches obtained from the two unrelated LK patients was used. Notably, all six revertant epidermis samples showed copy-neutral loss-of-heterozygosity (LOH) that started from regions centromeric to *LOR* and extended to the telomere of chromosome 1q, each with apparently different initiation sites for LOH (Fig 4), which excludes simple genetic mosaicism. In contrast, the affected skin did not show such LOH (Fig 4). These results clearly demonstrated that all revertants studied in the two LK families had resulted from independent somatic recombination.

### The mutant LOR protein causes disadvantages in cell survival

To analyse the pathogenicity of the mutant LOR protein, we generated Flp-In-293 cell lines stably expressing wild-type or the mutant. Although we observed little difference in cell viability between these cell lines (Fig 5A), wild-type cells displayed significantly higher proliferation rates and colony formation capacity than mutant cells (Fig 5B–E). These results indicate that the reversion of pathogenic mutations confers a fitness and survival advantage to keratinocytes, which leads to the emergence and expansion of visible revertant spots.

## Discussion

In this study, we clearly demonstrated that revertant mosaicism can occur in LK. Notably, long-term follow-up of our LK patients revealed the at least multi-year—and possibly lifetime—persistence of each revertant spot (Figs S4 and S5), which indicates that the reversion events must occur in long-lived epidermal SCs, not in short-lived progenitor cells whose in vivo half-life is 3–4 mo (Hirsch et al, 2017). In light of this, we propose two hypotheses, which are not mutually exclusive, for the frequent occurrence of recombination-mediated revertant spots in patients with LK. Our first hypothesis is that revertant keratinocytes are under positive selection. This explanation is supported by the fact that expression of the mutant LOR protein provides survival disadvantages to the Flp-In-293 cells (Fig 5). However, the clinically limited size of revertant spots, with a maximum diameter of 10 mm, also suggests that the growth of revertant clones eventually slows during their expansion, and the clonal selective advantage conferred by

reversion of the pathogenic mutations does not seem to continue as clones grow larger in LK. Notably, such a "constrained positive selection" phenomenon has also been observed in relation to cancer mutations (Martincorena & Campbell, 2015). In normal skin, despite positive selection and clonal expansion of epidermal cells that have acquired cancer driver mutations, the clone sizes are relatively limited and similar across individuals (Martincorena & Campbell, 2015). Addressing the mechanisms constraining the revertant clone expansion should represent a promising strategy to boost the potential of revertant cell therapies. Second, the production rate of revertant SC clones via somatic recombination may be elevated in LK. In inherited skin disorders with reports of revertant mosaicism, a combination of back mutation, second mutation, gene conversion, somatic recombination, and transcriptional slippage leads to gene reversion (Lim et al, 2017), with the exception of LK and IWC, in which somatic recombination is the only reported mechanism underlying the reversion of pathogenic mutations (Choate et al, 2010, 2015; Suzuki et al, 2016; Nomura et al, 2018). Furthermore, the number of revertant skin spots in LK and IWC is higher than that in other genodermatoses (Choate et al, 2010, 2015; Suzuki et al, 2016). These findings imply that an increased rate of somatic recombination underlies the high frequency of reversion events in LK and IWC. Notably, all LK cases and most, if not all, IWC cases carry pathogenic mutations that result in a frameshifted protein with an arginine-rich nuclear localization signal (Ishida-Yamamoto, 2003; Choate et al, 2010, 2015; Suzuki et al, 2016; Nomura et al, 2018). It has not escaped our notice that the positively charged arginine-rich frameshift mutant protein might cause frequent somatic recombination. Although an answer to the question of whether the frameshifted protein acts directly or indirectly in inducing recombination in epidermal SCs remains elusive, the lack of evidence for the expression of the terminal differentiation marker loricrin in those cells indicates an indirect effect.

The high accessibility of the skin is considered to facilitate the detection of revertant mosaicism. However, revertant mosaicism, as mentioned earlier, has never been reported in LK until now, and the following three possibilities might explain why it has gone undetected. First, the probands in the published case reports of LK, most of whom are under the age of 25, might be too young to exhibit noticeably sized revertant skin spots. Our long-term observation of two LK families reveals that age appears to be a key determinant for the development of revertant spots of recognizable size in patients with LK, as is the case with IWC (Suzuki et al, 2016). In support of this, recent studies have suggested that the accumulation of genetic mutations is directly associated with the frequency of cell division, which obviously increases with age (Tomasetti & Vogelstein, 2015), and that aging *Drosophila* intestinal SCs experience frequent somatic recombination that results in clonal mosaicism (Siudeja et al, 2015). Second, LK patients do not always present with erythema or erythematous plaques (Ishida-Yamamoto, 2003; Pohler et al, 2015); consequently, revertant skin spots easily escape patients' and even clinicians' attention. Indeed, in our cases, revertant skin spots were clinically detected as well-demarcated, round or oval-shaped islands of unaffected

(p.Gly183ArgfsTer153) and c.664_665insC (p.Gln222ProfsTer116) in family 1 and family 2, respectively. These mutations replace the C-terminus of LOR with novel arginine-rich peptides (shown in a different colour). The numerical values indicate the amino acid number of wild-type LOR from the C-terminus.

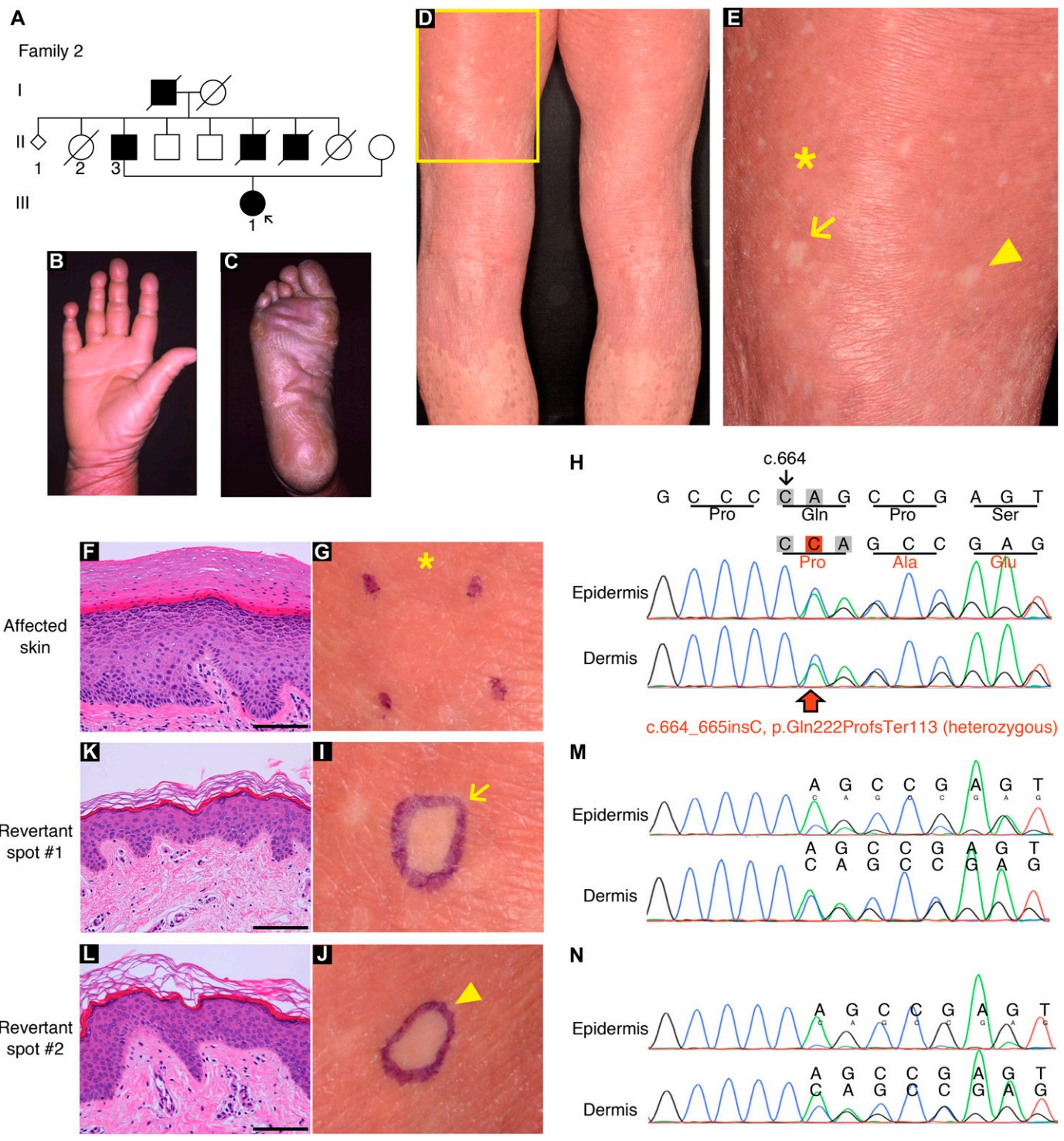

**Figure 3. Clinical, histological, and genetic features of the proband in LK family 2.**

**(A)** Pedigree. The arrow indicates the proband. **(B, C)** The proband showed PPK with a honeycomb pattern and constricted digits. These clinical images were taken when she was 39. **(D, E)** Generalized ichthyosis and well-demarcated erythematous plaques. She also presented with dozens of non-scaly whitish skin spots on her left thigh. Higher magnification of the square in (D) is shown in (E). Note that skin biopsies were taken from the affected lesion (asterisk) and two normal-appearing patches (arrow and arrowhead). **(F, G)** Histology of the lesional skin (G) showed marked hyperkeratosis with parakeratosis and hypergranulosis (F). **(H)** A heterozygous 1-bp insertion mutation, c.664_665insC (p.Gln222ProfsTer116), was detected in the affected epidermis and dermis. **(I–N)** The two normal-appearing patches (I, J) displayed normal epidermal architecture (K, L). The frameshift mutation was detected at lower levels in the revertant epidermis than in the revertant dermis, suggesting admixture between revertant and affected cells in the revertant epidermis (M, N). Scale bars, 100 μm.

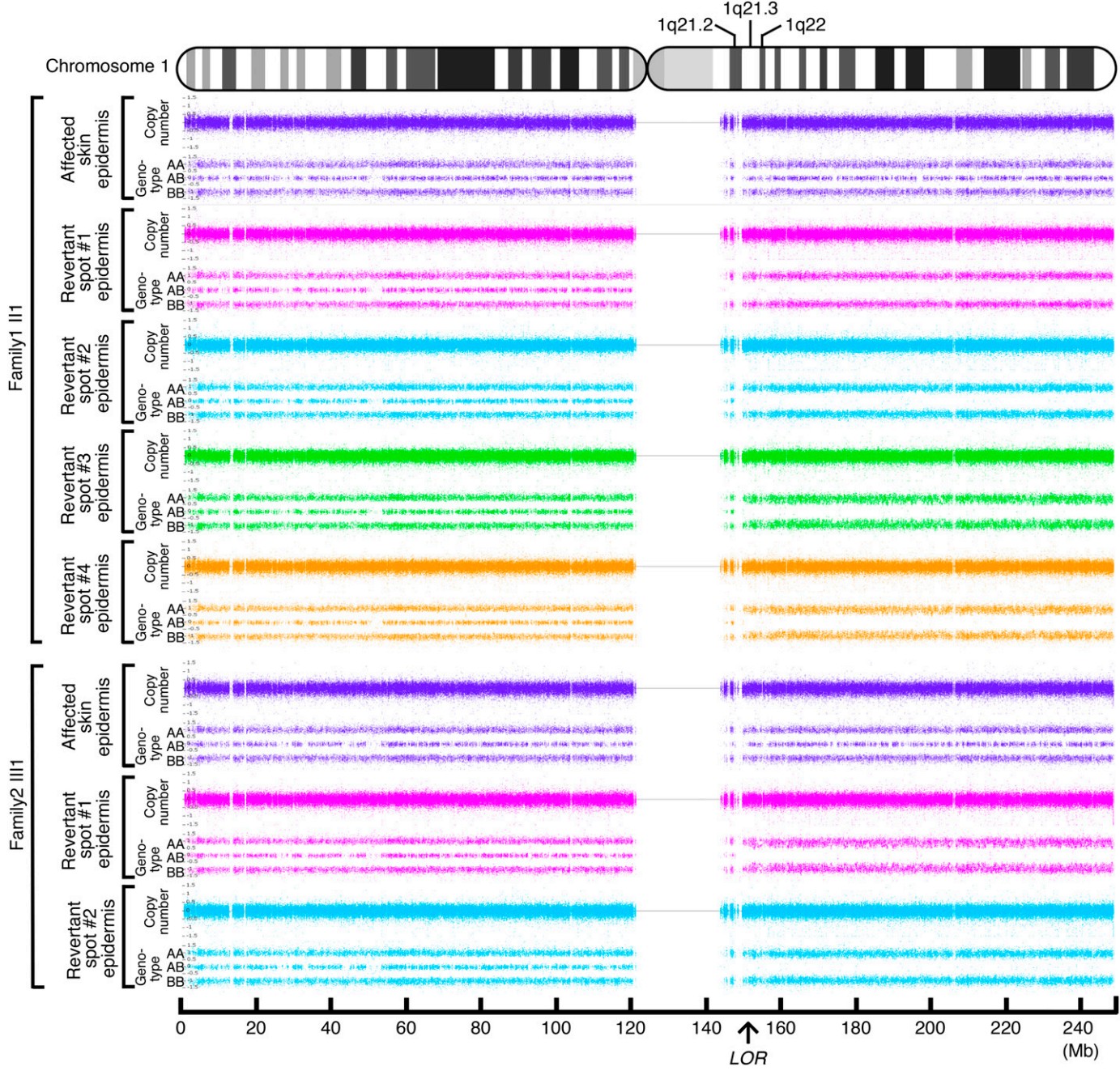

**Figure 4. Identification of LOH.**

Six revertant and two affected epidermal samples were analysed. Whole-genome single nucleotide polymorphism microarray analysis identified a copy-neutral LOH on 1q in all of the six revertant epidermis samples, whereas no LOH was detected in the two affected epidermis samples. The LOH began in regions centromeric to *LOR* and extended to the telomere of the chromosome. Note that no LOH was present on other chromosomes.

skin appearing on an erythematous background of affected skin (Figs 1I and 3E). Last, normal-coloured skin areas in LK patients do not always represent a revertant spot (Fig S7), which again hinders the detection of revertant mosaicism.

In conclusion, this study provides the first evidence for revertant mosaicism in LK. Furthermore, our results suggest that somatic recombination is the major, possibly sole, mechanism

responsible for the reversion of *LOR* mutations. The prominent self-healing phenomenon in LK suggests the possibility of manipulating somatic recombination to induce the reversion of disease-causing mutations. Future research into the molecular basis of revertant mosaicism might hold potential to benefit patients with LK and/or other currently intractable genetic diseases.

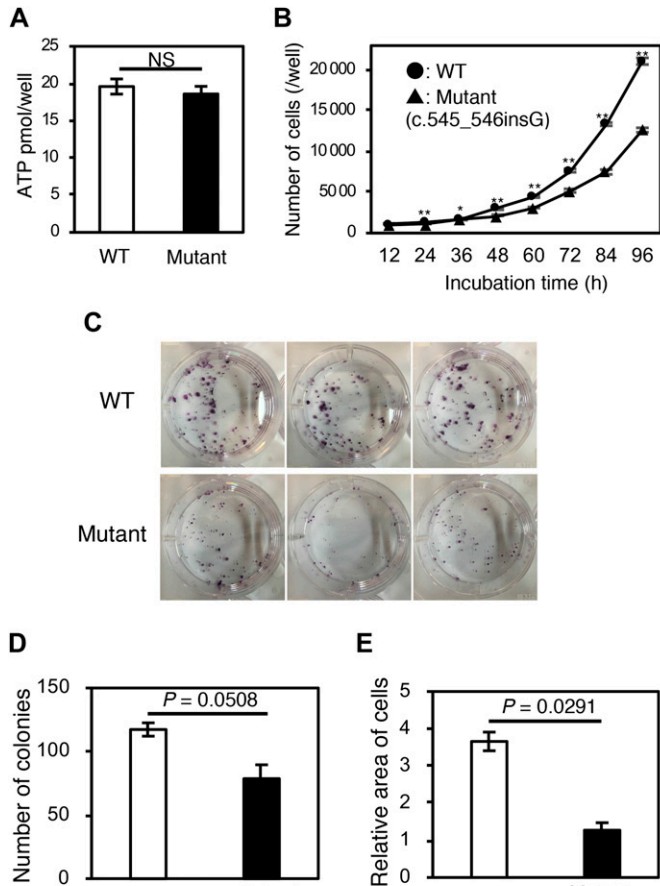

**Figure 5. Cell viability, cell proliferation, and colony formation assays.**
Flp-In-293 cell lines stably expressing wild-type or mutant LOR were analysed.
**(A)** The ATP-based cell viability assay revealed little difference in cell viability between these cell lines ($n = 6$). **(B)** Wild-type cells displayed significantly higher proliferation rates than mutant cells ($n = 6$). **(C–E)** Colony formation assay. Wild-type cells formed more colonies than mutant cells, albeit with a borderline statistical significance ($n = 3$). The area occupied by the wild-type colonies on the culture well surface was approximately 3 times larger than that occupied by the mutant colonies ($n = 3$). Data indicate mean ± SEM. *$P < 0.05$, **$P < 0.01$, $P = 0.0508$, $P = 0.0291$ by two-tailed $t$ test.

## Materials and Methods

### Genomic DNA extraction

For mutational analysis, genomic DNA of the patients and their family members was extracted from peripheral blood or saliva using a QIAamp DNA Blood Maxi kit (QIAGEN) or an Oragene DNA Self-Collection kit (DNA Genotek), respectively. Genomic DNA was also extracted from the probands' skin using a QIAamp DNA Mini kit (QIAGEN) after separation of the epidermis and dermis from punch-biopsied skin samples using ammonium thiocyanate solution, as described previously (Clemmensen et al, 2009).

### Whole-exome sequencing

Following the enrichment of the coding regions and intron/exon boundaries using the SureSelect Human All Exon V4 kit (Agilent

Technologies), whole-exome sequencing was performed on the HiSeq 2000 platform (Illumina) with 100-bp paired-end reads to a median coverage of 100×. The sequence data were analysed and annotated using DNAnexus software (DNAnexus Inc., https://www.dnanexus.com/, accessed May 2013). Among the identified variants, we focused primarily on coding variants, including nonsynonymous variants, splice-site variants, and insertions/deletions. Under the assumption of autosomal dominant inheritance with full penetrance, heterozygous variants that were present in all of the affected individuals but not in unaffected participants were selected as possible candidates, which included the frameshift mutation c.545_546insG in *LOR*.

### Sanger sequencing

The presence of disease-causing mutations in *LOR* was confirmed by the direct sequencing of PCR amplicons. Exon 2 of *LOR*, which contains the entire coding sequence of the gene, was amplified by PCR using AmpliTaq Gold PCR Master Mix (Thermo Fisher Scientific). The primer sequences for the PCR were F: 5′-GCAGCTGCCTCCG-TAAGTATC-3′ and R: 5′-AAGTGTGAATGAGCGAATGCAG-3′. The amplicons were treated with ExoSAP-IT reagent (Thermo Fisher Scientific), and sequencing reactions were performed using BigDye Terminator version 3.1 (Thermo Fisher Scientific) with the *LOR*-specific sequencing primer 5′-CAGGCGGTCCAGTGCCAG-3′. We obtained sequence data using an ABI 3130*xl* genetic analyser (Thermo Fisher Scientific), with standard run conditions for electrophoresis and data collection.

### Formalin-fixed, paraffin-embedded (FFPE) skin specimens and haematoxylin and eosin staining

Punch biopsy skin specimens from the patients were fixed with 10% neutral buffered formalin (pH 7.2) for 24 h at room temperature. Fixed skin specimens were dehydrated in ethanol followed by xylene and embedded in paraffin for 14 h using an automated tissue processor, Tissue-Tek VIP 6 (Sakura). FFPE skins were cut into 4 µm sections using an REM-710 microtome (Yamato Kohki). For haematoxylin and eosin staining, sections were deparaffinized by treatment with xylene and ethanol, hydrated, stained with haematoxylin followed by eosin and dehydrated through ethanol and xylene using a Tissue-Tek DRS 2000 automated staining system (Sakura).

### Immunohistochemistry

Sections were obtained from the FFPE skin specimens. To expose the antigens, deparaffinized and rehydrated sections were treated with 10 mM sodium citrate with 1 ml/l Tween 20 (Sigma-Aldrich), pH 6.0, for 10 min at 96°C and then cooled for 30 min at room temperature. After rinsing in PBS and blocking with 10% normal goat serum, sections were incubated with primary antibodies diluted in PBS followed by incubation with secondary antibodies conjugated to fluorescent dyes. The nuclei were counterstained by DAPI (Dojindo). Fluorescence images were obtained using an Olympus FV1000 confocal laser scanning microscope (Olympus).

## Antibodies

The primary antibodies used in this study included a rabbit anti-human LOR antibody (PRB-145P; Covance) for the wild-type LOR protein, a mouse anti-human KRT10 antibody (M7002; DAKO) for the visualization of differentiated keratinocytes, and a rabbit polyclonal antibody raised against synthetic mutant LOR-specific peptide VQIDPPGYH. The secondary antibodies used included an Alexa Fluor 488-conjugated goat anti-rabbit IgG (H + L) antibody (A11008; Thermo Fisher Scientific) and an Alexa Fluor 680-conjugated goat anti-mouse IgG (H + L) antibody (A21057; Thermo Fisher Scientific).

## Whole-genome oligo-single nucleotide polymorphism array

An Affymetrix CytoScan HD array (Thermo Fisher Scientific) containing more than 2.6 million markers was used to identify copy number variations and LOH using genomic DNA prepared from the epidermis as described previously (Nomura et al, 2018).

## Plasmids

Wild-type and mutant (c.545_546insG) LOR sequences were amplified from the genomic DNA of the proband in family 1 using PCR, as described above. The amplicons were cloned into the pCR-Blunt vector (Thermo Fisher Scientific) with the addition of the N-terminal 3×FLAG tag sequence. The sequence was subsequently confirmed by Sanger sequencing. For the Flp-In expression system (Thermo Fisher Scientific), the wild-type and mutant LOR sequences were subcloned individually into the pcDNA5/FRT mammalian expression vector (Thermo Fisher Scientific).

## Cell culture and transfection

Cells were cultured in DMEM (Wako) supplemented with 10% fetal bovine serum (Sigma-Aldrich) under standard cell culture conditions at 37°C and 5% $CO_2$. Transfection was performed using Lipofectamine 2000 (Thermo Fisher Scientific) according to the manufacturer's instructions.

## Flp-In expression system

Vectors expressing the wild-type or mutant LOR were transfected into the Flp-In-293 cell line (Thermo Fisher Scientific) with pOG44, a recombinase expression vector. After transfection, cells were cultured with 100 $\mu$g/ml hygromycin (Wako) for several passages to select the cells in which the expression cassettes were introduced into genomic DNA.

## Cell viability assay

Flp-In-293 cells expressing wild-type or mutant LOR were seeded in a 96-well plate (2.5 × 10$^3$ cells/well) and incubated for 24 h. Intracellular ATP concentrations (pmol/well) were determined using a luciferase-based luminescence assay, CellTiter-Glo 2.0 (Promega). Standard curves using known concentrations of ATP (Wako) were generated to calculate the ATP concentration in the cells.

## Cell proliferation assay

Flp-In-293 cells expressing wild-type or mutant LOR were seeded in a 96-well plate (2.5 × 10$^3$ cells/well) and incubated for 12, 24, 36, 48, 60, 72, 84, or 96 h. The cells were stained by adding 0.5 $\mu$g Hoechst 33342 (Dojindo) in 100 $\mu$l of 10% neutral buffered formalin (pH 7.2) to each well. The average cell number at each time point was measured using a BZ-700 fluorescence microscope (Keyence).

## Colony formation assay

Flp-In-293 cells expressing wild-type or mutant LOR were seeded in a six-well plate (5.0 × 10$^2$ cells/well) and incubated for 2 wk. The colonies were visualized by staining the cells with 0.5% crystal violet and 6% glutaraldehyde in PBS. The number and size of the colonies were measured using ImageQuant LAS 4000 (Fujifilm). Colony images were obtained using a common digital camera.

## Statistical analysis

Results of at least three biological replicates were represented as mean ± SEM. The significance of difference between two groups was analysed by $t$ test using Statplus Pro statistical analysis software (Analystsoft).

## Study approval

The patients and their family members provided written informed consent to participate in this study, in compliance with the Declaration of Helsinki. The Institutional Review Board at the Hokkaido University Graduate School of Medicine approved this study (project No. 14-063). Note that we do not have consent to share the raw whole-exome sequencing data.

# Supplementary Information

# Acknowledgements

We are most indebted to the patients and their family members for their participation in this study. This work was supported by the JSPS KAKENHI (grants JP15K09738 and JP17H06271 to T Nomura and H Shimizu, respectively), the Terumo Foundation for Life Sciences and Arts (grants 16-II 330 to T Nomura), the Rohto Dermatology Research Award (to T Nomura), the Akiyama Life Science Foundation (to T Nomura), the Nakatomi Foundation (to T Nomura), the Ichiro Kanehara Foundation (to T Nomura), and the Northern Advancement Center for Science & Technology Foundation (grant no. H28 T-1-42 to T Nomura).

## Author Contributions

S Suzuki: conceptualization, data curation, formal analysis, methodology, investigation, validation, writing—review and editing.
T Nomura: conceptualization, data curation, funding acquisition, project administration, resources, supervision, writing—original draft, review and editing.

T Miyauchi: resources, writing—review and editing.
M Takeda: methodology, writing—review and editing.
Y Fujita: resources, writing—review and editing.
W Nishie: resources, writing—review and editing.
M Akiyama: resources, writing—review and editing.
A Ishida-Yamamoto: resources, writing—review and editing.
H Shimizu: funding acquisition, writing—review and editing.

## Conflict of Interest Statement

The authors declare that they have no conflict of interest.

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
