## [Reviewer comments · Life Science Alliance]

Life Science Alliance

Somatic recombination underlies frequent revertant mosaicism in loricrin keratoderma

Shotaro Suzuki, Toshifumi Nomura, Toshinari Miyauchi, Masae Takeda, Yasuyuki Fujita, Wataru Nishie, Masashi Akiyama, Akemi Ishida-Yamamoto, and Hiroshi Shimizu

DOI: <https://doi.org/10.26508/lsa.201800284>

Corresponding author(s): Toshifumi Nomura, Hokkaido University Graduate School of Medicine

Review Timeline:

Submission Date:	2018-12-18
Editorial Decision:	2019-01-23
Revision Received:	2019-01-25
Accepted:	2019-01-28

Scientific Editor: Andrea Leibfried

Transaction Report:

January 23, 2019

RE: Life Science Alliance Manuscript #LSA-2018-00284-T

Dr. Toshifumi Nomura
Hokkaido University Graduate School of Medicine
Department of Dermatology
North 15 West 7, Kita-ku
Sapporo, Hokkaido 0608638
Japan

Dear Dr. Nomura,

Thank you for submitting your manuscript entitled "Somatic recombination underlies frequent revertant mosaicism in loricrin keratoderma" to Life Science Alliance. Your work was assessed by two reviewers and we enclose their reports below.

As you will see, the reviewers appreciate your data and only have a few suggestions on how to further strengthen your work. We would be thus happy to publish your paper in Life Science Alliance pending final revisions to address the comments raised by the reviewers as well as the following editorial points:

- please deposit the whole exome sequencing data in a repository (<https://ega-archive.org> or <https://www.ncbi.nlm.nih.gov/clinvar/>)
- please add the description for supplementary figure S6E to the legend
- please add the missing panel 'D' to supplementary figure S7
- please upload all supplementary figures as individual files

A. FINAL FILES:

-- High-resolution figure, supplementary figure and video files uploaded as individual files: See our detailed guidelines for preparing your production-ready images, <http://life-science-alliance.org/authorguide>

-- Summary blurb (enter in submission system): A short text summarizing in a single sentence the

study (max. 200 characters including spaces). This text is used in conjunction with the titles of papers, hence should be informative and complementary to the title. It should describe the context and significance of the findings for a general readership; it should be written in the present tense and refer to the work in the third person. Author names should not be mentioned.

B. MANUSCRIPT ORGANIZATION AND FORMATTING:

Full guidelines are available on our Instructions for Authors page, <http://life-science-alliance.org/authorguide>

Sincerely,

Andrea Leibfried, PhD
Executive Editor
Life Science Alliance
Meyershofstr. 1
69117 Heidelberg, Germany
t +49 6221 8891 502
e a.leibfried@life-science-alliance.org
www.life-science-alliance.org

Reviewer #1 (Comments to the Authors (Required)):

Suzuki et al. report on two unrelated families with lorocrin keratoderma where two different

pathogenic germline variants were identified in LOR. The probands were exhibiting dozens of clinically and histologically normalized skin areas in which the authors demonstrated that the causal variants were corrected. In the normalized skin spots, the authors also identified loss-of-heterozygosity on chromosome 1q from regions centromeric of LOR to the telomere. In contrast, the affected skin did not show any LOH. The authors therefore proposed that somatic recombination is the mechanism for the reversion of the LOR mutations. Based on in vitro studies the authors furthermore demonstrate that the reversion confers a growth and/or survival advantage to cells in vitro.

The authors demonstrate for the first time the occurrence of somatic revertant mosaicism in LOR. The data is solid and the manuscript is well written.

Minor comment

In the material and methods there is limited amount of information how the whole exome sequencing and analysis was performed. Was only the proband sequenced or were additional family members included in the analysis? Was in silico panels used? How was the filtering of variants performed?

Reviewer #2 (Comments to the Authors (Required)):

This is a well-written and illustrated account of the phenomenon of revertant mosaicism occurring in the setting of a skin disease, loricrin keratoderma, an autosomal dominant disorder in which heterozygous frameshift mutations in LOR usually result in an arginine rich tail to the protein, nuclear retention of the mutant protein, and the disruption to the cornification process in skin. The clinical, skin microscopy and functional data collectively provide new insight into this observation. Minor comments only.

[1] In the introduction, bottom of page 4, it would be helpful to mention the specific genes involved since there are at least 2 genes showing RM in junctional EB (COL17A1 and LAMB3) and two in IWC (KRT1 and KRT10).

[2] Perhaps the extrapolation to therapy for patients should be downplayed a bit (abstract and discussion). Thus far, the most successful translation (with patient benefit) has been punch grafting for LAMB3 reported 5 years ago.

RE: MS# LSA-2018-00284-T

Point-by-point responses to the comments from the editor and reviewers

Editor:

Comments:

Please deposit the whole exome sequencing data in a repository. Please add the description for supplementary figure S6E to the legend. Please add the missing panel 'D' to supplementary figure S7. Please upload all supplementary figures as individual files.

Answer:

In accordance with the editor's suggestion, we have (1) modified the legend for Figure S6 to add the description for Figure S6E, (2) added Figure S7D, and (3) uploaded all supplementary figures as individual files. Unfortunately, we failed to obtain informed consent from the patient to deposit the whole exome sequencing data in a public repository, although he agreed with us to publish his clinical data including his genetic results in scientific journals. We have also reformatted our manuscript to meet the journal's requirements for a research article.

Reviewer #1:

Suzuki et al. report on two unrelated families with lorycin keratoderma where two different pathogenic germline variants were identified in LOR. The probands were exhibiting dozens of clinically and histologically normalized skin areas in which the authors demonstrated that the causal variants were corrected. In the normalized skin spots, the authors also identified loss-of-heterozygosity on chromosome 1q from regions centrometic of LOR to the telomere. In contrast, the affected skin did not show any LOH. The authors therefore proposed that somatic recombination is the mechanism for the reversion of the LOR mutations. Based on in vitro studies the authors furthermore demonstrate that the reversion confers a growth and/or survival advantage to cells in vitro. The authors demonstrate for the first time the occurrence of somatic revertant mosaicism in LOR. The data is solid and the manuscript is well written.

Comment #1:

In the material and methods there is limited amount of information how the whole exome sequencing and analysis was performed. Was only the proband sequenced or were

additional family members included in the analysis? Was in silico panels used? How was the filtering of variants performed?

Answer:

We thank the reviewer for the positive and supportive comments. We performed whole exome sequencing in 3 affected and 2 unaffected individuals in the family 1 without using *in silico* panels. Because the diagnosis of loricerin keratoderma was clinically and histologically suspected, we could easily identify the pathogenic mutation in *LOR* in this family. The revised manuscript now mentions these points and the filtering strategy used in this study as follows:

Results, 1st paragraph

“Whole-exome and Sanger sequencing in 3 affected and 2 unaffected individuals in the family revealed that the affected individuals were all heterozygous for a 1-bp insertion mutation in *LOR*, c.545_546insG (p.Gly183ArgfsTer153) (Fig 1G), which had previously been reported to cause LK (Song *et al*, 2008), further verifying the diagnosis of LK. By contrast, the unaffected individuals were both wild-type for the mutation.”

Materials and Methods, “Whole-exome sequencing”

“Following the enrichment of coding regions and intron/exon boundaries using the SureSelect Human All Exon V4 Kit (Agilent Technologies), whole-exome sequencing was performed on the HiSeq 2000 platform (Illumina) with 100-bp paired-end reads to a median coverage of 100×. The sequence data were analysed and annotated using DNAnexus software (DNAnexus Inc., <https://www.dnanexus.com/>, accessed May 2013). Among the identified variants, we focused primarily on coding variants, including nonsynonymous variants, splice-site variants and insertions/deletions. Under the assumption of autosomal dominant inheritance with full penetrance, heterozygous variants that were present in all of the affected individuals but not in unaffected participants were selected as possible candidates, which included the frameshift mutation c.545_546insG in *LOR*.”

Reviewer #2:

This is a well-written and illustrated account of the phenomenon of revertant mosaicism occurring in the setting of a skin disease, loricerin keratoderma, an autosomal dominant

disorder in which heterozygous frameshift mutations in LOR usually result in an arginine rich tail to the protein, nuclear retention of the mutant protein, and the disruption to the cornification process in skin. The clinical, skin microscopy and functional data collectively provide new insight into this observation.

Comment #1:

In the introduction, bottom of page 4, it would be helpful to mention the specific genes involved since there are at least 2 genes showing RM in junctional EB (COL17A1 and LAMB3) and two in IWC (KRT1 and KRT10).

Answer:

We thank the reviewer for these insightful comments. In accordance with the reviewer's suggestion, we have revised the manuscript as shown below.

Introduction, 1st paragraph

“However, this clinically important ‘natural gene therapy’ phenomenon has been documented in only ~30 diseases, including 7 genodermatoses caused by mutations in COL7A1 (recessive dystrophic epidermolysis bullosa [EB, MIM: 226600]), COL17A1 or LAMB3 (junctional EB [MIM: 226650]), KRT14 (EB simplex [MIM: 131900]), FERMT1 (Kindler syndrome [MIM: 173650]), KRT10 or KRT1 (ichthyosis with confetti [IWC, MIM: 609165]) and GJB2 (keratitis-ichthyosis-deafness syndrome [MIM: 148210]) (Gudmundsson et al, 2017; Jonkman & Pasmooij, 2009; Lim *et al*, 2017; van den Akker *et al*, 2018).”

Comment #2:

Perhaps the extrapolation to therapy for patients should be downplayed a bit (abstract and discussion). Thus far, the most successful translation (with patient benefit) has been punch grafting for LAMB3 reported 5 years ago.

Answer:

We fully understand the reviewer's concern. In accordance with the reviewer's suggestion, we have revised the manuscript as shown below.

Abstract

“Nevertheless, the identification of revertant mosaicism in LK might pave the way for

revertant therapy for this intractable disease.”

Discussion, 3rd paragraph

“Future research into the molecular basis of revertant mosaicism might hold potential to benefit patients with LK and/or other currently intractable genetic diseases.”

January 28, 2019

RE: Life Science Alliance Manuscript #LSA-2018-00284-TR

Dr. Toshifumi Nomura
Hokkaido University Graduate School of Medicine
Department of Dermatology
North 15 West 7, Kita-ku
Sapporo, Hokkaido 0608638
Japan

Dear Dr. Nomura,

Thank you for submitting your Research Article entitled "Somatic recombination underlies frequent revertant mosaicism in loricrin keratoderma". We appreciate the introduced changes and it is a pleasure to let you know that your manuscript is now accepted for publication in Life Science Alliance. Congratulations on this interesting work.

DISTRIBUTION OF MATERIALS:

Again, congratulations on a very nice paper. I hope you found the review process to be constructive and are pleased with how the manuscript was handled editorially. We look forward to future exciting submissions from your lab.

Sincerely,
